# Attenuation in Nicotinic Acetylcholine Receptor α9 and α10 Subunit Double Knock-Out Mice of Experimental Autoimmune Encephalomyelitis

**DOI:** 10.3390/biom9120827

**Published:** 2019-12-04

**Authors:** Qiang Liu, Minshu Li, Paul Whiteaker, Fu-Dong Shi, Barbara J. Morley, Ronald J. Lukas

**Affiliations:** 1Division of Neurobiology, Barrow Neurological Institute, Phoenix, AZ 85013, USA; qiang.liu@dignityhealth.org (Q.L.); minshuli2012@163.com (M.L.); paul.whiteaker@dignityhealth.org (P.W.); fshi66@gmail.com (F.-D.S.); 2Boys Town National Research Hospital, Omaha, NE 68131, USA

**Keywords:** autoimmunity, cholinergic anti-inflammatory pathway, experimental autoimmune encephalomyelitis, inflammation, multiple sclerosis, nicotinic acetylcholine receptors

## Abstract

Experimental autoimmune encephalomyelitis (EAE) is attenuated in nicotinic acetylcholine receptor (nAChR) α9 subunit knock-out (α9 KO) mice. However, protection is incomplete, raising questions about roles for related, nAChR α10 subunits in ionotropic or recently-revealed metabotropic contributions to effects. Here, we demonstrate reduced EAE severity and delayed onset of disease signs in nAChR α9/α10 subunit double knock-out (DKO) animals relative to effects in wild-type (WT) control mice. These effects are indistinguishable from contemporaneously-observed effects in nicotine-treated WT or in α9 KO mice. Immune cell infiltration into the spinal cord and brain, reactive oxygen species levels in vivo, and demyelination, mostly in the spinal cord, are reduced in DKO mice. Disease severity is not altered relative to WT controls in mice harboring a gain-of-function mutation in α9 subunits. These findings minimize the likelihood that additional deletion of nAChR α10 subunits impacts disease differently than α9 KO alone, whether through ionotropic, metabotropic, or alternative mechanisms. Moreover, our results provide further evidence of disease-exacerbating roles for nAChR containing α9 subunits (α9*-nAChR) in EAE inflammatory and autoimmune responses. This supports our hypothesis that α9*-nAChR or their downstream mediators are attractive targets for attenuation of inflammation and autoimmunity.

## 1. Introduction

Nicotinic acetylcholine receptors (nAChR) are prototypical members of the cys-loop superfamily of neurotransmitter-gated ion channels [1,2,3]. These entities assemble as homo- or hetero-pentameric combinations of subunits generated from a family of sixteen different genes in mammals, with some genes being alternatively spliced, clustered, or duplicated, sometimes in species-specific ways [1,2,3]. nAChR exist as a group of subtypes that are classified according to their subunit compositions; an “*” is used to indicate a set of nAChR subtypes containing the specified subunits, but also to indicate that subunits in addition to those specified may or are known to be partners in the receptor assembly [4]. As their names imply, all nAChR respond to the natural neurotransmitter and chemical signaling agent, acetylcholine (ACh), and to one of the biologically-active agents in tobacco or other plants, nicotine, which exerts its actions on the brain and body via nAChR. 

ACh has specialized activity as a neurotransmitter in the nervous system, but it also has actions, even early in development, in non-neuronal tissues and organs, such as the immune system [5,6]. Immune system cells express nAChR subtypes and subunits and enzymes that synthesize or degrade ACh [7,8,9,10,11,12,13,14]. Types and levels of nAChR subtype and subunit expression are dependent on developmental stage and immune system activation state [9,15,16,17]. Thus, everything needed is present for immune system cholinergic signaling, whether via paracrine- or autocrine-type mechanisms, and regardless of whether ACh signaling also is provided via fine innervation of immune system organs.

Maintenance of health via protection against infectious agents and consequences of cell or tissue damage is dependent on normally functioning inflammatory and immune responses [18,19,20,21]. However, many chronic and debilitating, personally and economically costly diseases are caused or exacerbated by aberrant inflammatory and immune responses [18,19,20,21,22,23]. For example, excessive or protracted inflammatory or immune responses, especially upon transitioning to autoimmune (anti-self) activity, contribute to or cause diseases [24]. Moreover, an imbalance in protective compared to disease-exacerbating immune system and inflammatory responses has been implicated in disorders such as multiple sclerosis (MS), Alzheimer’s or Parkinson’s diseases, autism, stroke, and spinal cord or brain injury, or in cancer centrally or peripherally [19,21,23,25,26,27,28,29]. Clinical advances promoting disease-attenuating influences and inhibiting disease-exacerbating effects require an improved understanding of immune and inflammatory processes.

Prompted by our earlier findings that nicotine has immunosuppressive effects, mediated in part by inhibition of T cell differentiation and development [16], we hypothesized that nicotine-mediated immunosuppression could be leveraged to advantage in an autoimmune disease context. We and others then found that nicotine exposure suppresses inflammatory and immune responses in the experimental autoimmune encephalomyelitis (EAE) model of some forms of MS [15,30,31]. We also found that there is abundant expression in almost every immune cell type investigated of α9 subunits that make up α9*-nAChR [15,17]. Notably, and of relevance to interpretation of results in this and our prior studies, nicotine acts as an antagonist of α9*-nAChR ion channel function, rather than an agonist as it is for other ionotropic nAChR subtypes [1,2,3]. Next, we showed that protective effects of nicotine exposure on the EAE response are mimicked by deletion of nAChR α9 subunits in α9 subunit knock-out (α9 KO) animals [17]. That is, nicotine’s inhibition of α9*-nAChR function, or genetic deletion of those receptors, are equally protective against EAE. Furthermore, we also demonstrated using adoptive transfer experiments that genetic deletion of nAChR α9 subunits in donor mouse peripheral immune cells is adequate to provide maximal protection against EAE [32], consistent as well with the consensus that there is no expression of α9*-nAChR in the healthy mammalian brain [33,34]. Nevertheless, whether EAE is induced directly, or by adoptive transfer, nAChR α9 subunit deletion does not confer complete protection against EAE. This raises questions about other possible participants in EAE effects and whether greater or less protection might be conferred by other nAChR manipulations.

nAChR α10 subunits are known assembly partners for α9 subunits, and their co-expression dramatically elevates ionotropic function of heteromeric α9α10-nAChR compared to that of homomeric α9-nAChR [2,33]. Homomeric α9-nAChR that function as ACh-gated ion channels at discrete sites on immune cell membranes would allow Ca^2+^ entry, which in turn could coordinate with, for example, T cell receptor-mediated changes in intracellular Ca^2+^ signaling to affect cell function [35]. Such function would be elevated when there is expression by immune cells of heteromeric α9α10-nAChR. Moreover, although there is no evidence that α10 subunits alone can form ionotropic nAChR, there is a formal possibility that they could assemble with nAChR subunits other than α9 to form functional ion channels. Furthermore, of relevance is evidence for metabotropic function of nAChR α9 or α10 subunits [36,37,38]. Additionally, ligand occupancy of subunits or assembled receptors could simply alter nAChR scaffolds for interactions with other intracellular signaling molecules or cytoskeletal elements, even to the point of affecting gene expression [2,39,40]. Whether via such unconventional means, or via ionotropic or metabotropic mechanisms, primordial nAChR α9 or α10 subunits or receptors could act to affect immune cell function [2,35,41,42].

We sought to address these possibilities and to further define roles for nAChR in inflammatory and autoimmune disease pathogenesis, recognizing that nAChR α10 subunits also are expressed by some immune cells [8,43,44]. We determined whether attenuation of EAE disease signs, demyelination, and inflammation seen for EAE in α9 KO or nicotine-treated mice is retained, enhanced, or reversed in animals having additional elimination of α10 subunits. Additional studies were done to determine whether augmentation of α9*-nAChR ionotropic function via introduction of a gain-of-function (GOF) mutant in an α9 GOF knock-in (KI) mouse produces an altered EAE disease profile.

## 2. Materials and Methods

### 2.1. Mice

C57BL/6J mice were sometimes purchased from Taconic Biosciences (Rensselaer, NY, USA) as wild-type (WT) control animals, but most studies used WT littermates or colony controls obtained in breeding to derive mice in which there was constitutive α9 KO or α9/α10 DKO, all extensively backcrossed to the C57BL/6J background. Disease severity in WT animals from Taconic or our breeding colony was indistinguishable for animals treated with the same lots of inoculant and pertussis toxin (see below).

Details about generation of nAChR α9 and α10 subunit KO mice have been described previously [45,46]. Briefly, the α9 KO construct involved deletion of exons 1 and 2 plus flanking intronic sequences, and the α10 KO construct involved deletion of exons 1–3 plus flanking introns (Genoway, Inc., Lyon, France). Embryonic stem cells used to make transgenic mouse lines were from the 129/svPas (129S2; 129/sv) strain. Both KO lines were extensively backcrossed at the Boys Town National Research Hospital (Omaha, NE, USA) to C57BL/6J mice (Jackson Laboratory, Bar Harbor, ME) using marker-assisted accelerated backcrossing (MAX BAX; Charles River, Troy, NY, USA) until congenicity (99+%) was achieved. There is absence in the relevant KO mice of α9 transcript in cells dissected from the retina and inner ear [45,46,47] and of α10 transcript in the inner ear [45,46], i.e., at sites where they are naturally expressed.

The nAChR α9/α10 subunit DKO mouse model was generated by crossing heterozygous or homozygous α9 KO female mice with heterozygotic or homozygotic α10 KO male mice. Animals used for the studies reported here were bred, housed, and genotyped at Boys Town National Research Hospital under identical conditions to optimize quality control and then shipped at weaning before being housed at animal facilities of the Barrow Neurological Institute (Phoenix, AZ, USA). Litter sizes for all strains varied from 5–9. Dams and males are typically first bred at 6–8 weeks of age, not utilized for more than two litters, and are rarely bred past the age of 12–14 weeks, all to reduce the incidence of epigenetic factors that might affect phenotype. Animals are periodically backcrossed to C57BL/6J mice to prevent genetic drift from the background strain. Animals were negative for all standard viruses and parasites (IDEXX, Columbia, MO, USA). All animal studies were approved by Institutional Animal Care and Use Committees (IACUC) of the Barrow Neurological Institute (A3519-01, Protocol 519) or Boys Town National Research Hospital (A3349-01, Protocol 16-04).

nAChR α9 subunit GOF KI mice having a leucine-to-threonine substitution at the “9′” position in the second transmembrane domain of the subunit and thought to contribute to the ion channel gate were kindly provided by Dr. Paul Fuchs (John Hopkins University School of Medicine, Baltimore, MD, USA), as approved by his organization’s IACUC, and were generated as described [48]. They are on the FVB.129P2-*Pde6b^+^ Tyr^c-ch^*/AntJ (FVB; stock no. 004828, Jackson Laboratory, Bar Harbor, ME) background, purportedly to avoid late onset cochlear degeneration and hearing loss found in C57BL/6J mice, with compensation for retinal degeneration and blindness characteristic of FBV mice thanks to expression of the Pde6b allele [48]. α9 KI mice used for the studies reported here were genotyped and shipped at weaning before being housed in the Barrow Neurological Institute vivarium.

### 2.2. EAE Induction

EAE was induced in mice by hind flank sub-cutaneous injection with mouse myelin oligodendrocyte glycoprotein peptide_35–55_ (MOG; amino acids 35-MEVGWYRSPFSRVVHLYRNGK-55; purity > 95%; Bio-Synthesis, Inc.; Lewisville, TX, USA) in complete Freund’s adjuvant (Becton Dickenson Biosciences, Franklin Lakes, NJ, USA) containing 500 µg of nonviable desiccated Mycobacterium tuberculosis [31]. Mice also were injected intravenously with 200 ng pertussis toxin (List Biological Laboratories, Campbell, CA, USA) on the day of and on the second day after MOG injection [31]. Symptoms on a standardized scale from zero to five of disease severity (briefly: 0, no symptoms; 1, flaccid tail; 2, hindlimb weakness or abnormal gait; 3, complete hindlimb paralysis; 4, complete hindlimb paralysis with forelimb weakness or paralysis; 5, moribund or deceased) were monitored and scored daily as previously described [17,31]. Importantly, we carefully replicated, and modulated as necessary, our previously-published protocols (mouse age, gender, inoculation conditions, reagent parameters) to provoke peak disease severity in WT animals used as “positive controls” in this study that is very similar to peak disease severity in WT mice as achieved in our other work, some of which was done quite contemporaneously. This approach permits results to be compared across experiments using different cohorts of “experimental” or “test” animals, certainly qualitatively, but also with good confidence quantitatively. That is, WT animals represent a “100% disease” standard/“positive control” (analogous to “total radioligand binding”), and we know that no disease is present in non-inoculated animals, which thus represent a “0% disease” standard/”negative control” (analogous to “non-specific radioligand binding”). Experimental animals have intermediate levels of disease, which can be compared across flights of animals subjected to EAE when expressed as a percent of control disease levels, or even in absolute terms when disease severity on the standardized scale in WT animals is quantitatively similar across cohorts and different flights of animals. Our approach also means that peak disease severity attained is not so high as to be at a ceiling level and not so low as to be at a floor level, thus allowing room to observe any disease attenuation or exacerbation beyond what we have seen previously in α9 KO or nicotine-treated animals. This also allowed us to meet additional objectives of minimizing animal use, avoiding needless replication of prior findings, and bringing the studies to a manageable scale.

### 2.3. In Vivo Central Nervous System Bioluminescence

Reactive oxygen species (ROS) production in brain was ascertained as photons/(sec-10^5^) in regions of interest (corpus callosum) using bioluminescence images captured in live mice using the Xenogen IVIS200 imager and Living Image^®^ software (Caliper Life Sciences, Hopkinton, MA, USA) 20 min after intra-peritoneal injection of 100 µL of 50 mg/mL Luminol (Sigma-Aldrich, St. Louis, MO, USA), as previously described [17,49]. Any ROS production in non-immunized, WT, or transgenic animals maintained under pathogen-free conditions is below limits of detection, as for uninvolved brain or spinal cord regions in animals displaying EAE.

### 2.4. Histological Analyses

Mice selected for assessment because they exhibited disease severity levels representative of those for their group were anesthetized by 2% isoflurane inhalation at specific times after EAE induction and subjected to intracardiac perfusion with 50 mL of cold phosphate-buffered saline before spinal cords and brains were isolated and fixed in 10% formalin in phosphate-buffered saline. Longitudinal sections of the cervical enlargement or of the corpus callosum were embedded in paraffin and stained for infiltrating immune cells (hematoxylin and eosin (H&E)) or myelin (luxol fast blue). Degrees of inflammatory cell infiltration or demyelination were assessed manually on a five-point scale (e.g., for myelin status; 0, no changes; 1, focal area involvement; 2, <5% of total myelin area involved; 3, 5–10% of total myelin area involved; 4, 10–20% involved area; 5, >20% of total myelin area involved [50]). For histological analyses, four sections per mouse were examined, and results were quantified as averages from a total of n = 12 sections of N = 3 mice.

### 2.5. Statistical Analyses

Data are presented as means ± S.E.M. in Figure 1, Figure 2, Figure 3, Figure 4 and Figure 5. Differences were considered to be statistically significant at *p* < 0.05, and the studies were done with numbers of animals typical for such investigations and adequate to support declarations of substantial biological effects. Statistical differences among groups were evaluated by Mann–Whitney test for Figure 1; Figure 5 [51], by two-tailed unpaired Student’s *t*-test for Figure 2, Figure 3 and Figure 4, or by one-way ANOVA followed by Tukey post-hoc test for Table 1; Table 2. All statistical analyses were performed using Prism 5.0 software (GraphPad, San Diego, CA, USA). Group sizes used in this study are consistent with those employed across multiple, peer-reviewed, published studies and have been proven to be sufficient to produce statistically-significant outcomes of nAChR-directed manipulations in each case, as they have been in this study for adequately large effect sizes observed.

## 3. Results

### 3.1. Attenuated EAE Disease Severity in α9/α10 DKO Mice

In our earlier work, we demonstrated that there is delayed onset and diminished severity of EAE in nAChR α9 subunit KO animals when compared to WT littermates [17]. Although these protective effects are clearly evident, there is incomplete elimination of EAE in α9 KO mice. To address the possibility that nAChR α10 subunits engage in any of several mechanisms to affect autoimmunity (see Introduction), studies were done with mice lacking both nAChR α9 and α10 subunits, monitoring their EAE response following mouse MOG inoculation. Importantly, well-established disease provocation conditions were modulated so that peak disease severity for WT animals used as positive controls in this study was very similar to that for WT controls used in our earlier studies, and at a level that avoids ceiling or floor effects, this allowing for results to be compared across those studies (see Materials and Methods). MOG-immunized mice typically develop a maximum level of EAE followed by symptomatic improvement when neuroinflammation wanes [51], but most MOG-immunized mice retain some degree of neurodeficit. Relative to the time course for development of EAE and its severity in WT C57BL/6J mice, there is reduced peak disease severity in α9/α10 DKO animals (2.7 ± 0.3 for WT, but 1.6 ± 0.2 for DKO animals; *p* = 0.0003, Mann–Whitney test; Figure 1; see also Table 1 for additional assessments of mean disease severity score at day 15 and for cumulative disease scores and statistical comparisons). There is delayed onset of disease signs in α9/α10 DKO mice (14 days until first animal evidence of disease and 14.6 days average onset for α9/α10 DKO mice; eight days for first animal onset of disease signs and average onset at 9.6 days for WT mice). The time when there is a peak in disease severity also is later in α9/α10 DKO animals (day 20) than in WT littermate controls (day 15), and percent recovery from peak disease is better in α9/α10 DKO mice (75%) than for WT littermates (46%; disease severity at day 30 compared to disease severity on the day of peak disease; *p* < 0.05). Protection against EAE again is incomplete (i.e., even in DKO animals, there are disease signs). However, the magnitude of reduction in disease severity relative to that seen in WT controls and the delay in disease onset and peak disease expression are indistinguishable from effects seen for α9 KO alone or for nicotine treatment of WT animals, all on the same background (compare Figure 1 to [17]). That is, the average time of disease onset in previously-published studies was 8.7 days for WT animals, 14.5 days for α9 KO mice, and 15.7 days for nicotine-treated mice [17]. Peak disease severity was 2.7, 1.5, and 1.3, respectively, for those cohorts [17].

### 3.2. Diminished Inflammatory Indices in EAE α9/α10 DKO Mice

H&E staining for infiltrating immune cells was done at peak disease, 14 days after MOG inoculation, to assess levels of immune cell infiltration into the brains (corpus callosum) and spinal cords (cervical enlargement) of EAE mice. Results reveal that there is reduced (~40%) inflammatory cell presence in both central nervous system domains in α9/α10 DKO mice compared to levels in WT littermates (1330 ± 180 cells/mm^2^ in WT versus 820 ± 130 in DKO animal spinal cords, *p* = 0.008; 1080 ± 190 cells/mm^2^ in WT versus 610 ± 160 in DKO mouse brains, *p* = 0.02; Figure 2). We previously observed that nicotine exposure, which has effects indistinguishable from, and neither synergizing with nor counteracting, those in α9 KO animals, induced an about 70% reduction of immune cell infiltration into the spinal cord, measured 25 days after inoculation [31].

Similarly, measures were done of ROS imaged in vivo in the brain area of interest (corpus callosum) and defined by Luminol emission levels [illustrated here for animals at disease peak for WT mice, 14 days post-EAE induction; quantified as p/(sec-10^5^)]. ROS levels also are reduced, by about 30%, in α9/α10 DKO mice (1.5 ± 0.3) relative to WT animals (2.4 ± 0.4; *p* = 0.04; Figure 3). A similar degree of ROS reduction (25–50%) was seen before in the brains of animals subjected to nicotine treatment and also measured 14 days after inoculation [15,32].

### 3.3. Demyelination is Reduced in EAE α9/α10 DKO Mice

Relative to the extent of spinal cord demyelination observed in WT mice as they develop EAE, there is a ~50% attenuation of myelin loss in α9/α10 DKO mouse spinal cords 30 days after induction of EAE (compared to no demyelination in non-diseased WT mice, there is 38% ± 6% demyelination in WT mice in which EAE was produced versus 20% ± 8% demyelination in EAE-DKO mice; *p* = 0.007; Figure 4). In the animals studied, overall levels of brain demyelination were substantially lower than in the spinal cord (Figure 4). This was the case for WT mice (10% ± 2% myelin loss compared to non-disease controls) or for α9/α10 DKO animals (8.6% ± 2.3% myelin loss; *p* = 0.2, Figure 4). These findings indicate that there was little-if-any protection against brain demyelination in α9/α10 DKO mice. We previously showed that nicotine exposure similarly reduced fast blue staining, reflecting spinal cord demyelination, relative to that in WT mice treated instead with the vehicle, by about 65% measured 25 days after inoculation [31].

### 3.4. Effects of a nAChR α9 Subunit Gain-of-Function Mutation

Some nAChR subtypes display a GOF when certain mutations are introduced into subunit residues thought to line the nAChR ion channel [52]. Among the effects observed are increases in agonist potency/sensitivity, decreased desensitization (i.e., decreased loss of functional response upon sustained exposure to agonist), and/or changes in nature of a ligand from an antagonist to a partial agonist or from a partial to a fully efficacious agonist [52]. Such a mutation (leucine-to-threonine) was introduced into a second transmembrane domain residue thought to contribute to the ion channel gate of the rat nAChR α9 subunit [53]. In vitro, i.e., when expressed heterologously in Xenopus oocytes, these rat nAChR α9^L9′T^ subunits confer to α9-nAChR a 2.7-fold greater sensitivity to ACh and a lower rate of desensitization, and similar effects are seen for ionotropic function of α9α10-nAChR [53]. Mice expressing α9 subunits harboring the analogous L9′T mutation instead of the native subunit were created using a KI strategy [48]. In these mice, consistent with GOF effects observed for rat α9*-nAChR in vitro, there is an increase in sensitivity of inner and outer cochlear hair cell electrophysiological responses to ACh, slowed desensitization, and increased synaptic efficacy, revealing an auditory system GOF phenotype [48]. Here, these nAChR α9 subunit GOF/KI mice were used in a set of studies designed to determine whether elevation in α9*-nAChR ion channel function would alter EAE disease course and severity, just as might occur for increased ratios of ionotropic α9α10-nAChR to homomeric, ionotropic α9-nAChR.

In our initial studies, we used the same C57BL/6J WT mice that had served as controls for our other investigations, and we found that homozygous nAChR α9 KI mice on the FVB background have markedly lower disease responses to EAE induction. Since we had reasoned that α9 subunit GOF/KI mice might instead have abnormally higher autoimmune and inflammatory responses, we checked premises and repeated the study, again using stock, C57BL/6J WT mice as one set of controls, but also employing WT and α9 KI heterozygous littermates in addition to α9 KI homozygotic animals on the FVB background. EAE disease courses for all four cohorts were qualitatively comparable, with initial animal disease onset at day 10–11 (average days until onset of 11.0–11.9 across the cohorts) and peak disease at day 15–16, all followed by recovery. Results were informative, in that FVB WT control littermates also had markedly reduced EAE average peak disease severity relative to C57BL/6J WT controls (0.8 ± 0.1 for FVB WT versus 2.4 ± 0.3 for C57BL/6J WT; *p* < 0.0001; Figure 5), as did heterozygous (1.0 ± 0.1; FVB α9 KI HET versus C57BL/6J WT: *p* = 0.004) or homozygous (0.8 ± 0.3; FVB α9 KI HOMO versus C57BL/6J WT: *p* = 0.0004) α9 KI mice. This indicated that the much lower disease severity in the FVB α9 GOF/KI mice cohorts underscores the strong influence of genetic background on biological responses to insults. That aside, across the animals on the FVB background, there was no statistically-significant difference in peak disease severity between WT littermates or α9 GOF/KI hetero- or homo-zygotes (Figure 5; Table 2). However, there was a not-statistically-significant trend toward faster and more complete recovery from EAE for FVB WT littermates than for hetero- or homo-zygotes (82% versus 57% and 47%, respectively; 58% for C57BL/6J WT mice).

## 4. Discussion

nAChR expressed in the immune and nervous systems afford many mechanisms for modulation of neuroimmune processes [15,17,31,32]. We previously showed attenuation of EAE in α9 KO animals [15,17,31,32]. This suggested roles for α9*-nAChR in disease pathogenesis in an MS model. However, protection is incomplete, leading to hypotheses about possible compensatory or α9*-nAChR-independent mechanisms in α9 KO animals. nAChR α10 subunits are assembly partners in creating more highly functional, ionotropic α9α10-nAChR [2,33], and they also seem capable of functioning alone in a metabotropic capacity [36,37,38]. This study exploited both α9/α10 DKO mice and an α9*-nAChR GOF mouse line to assess possible ionotropic, metabotropic, or other roles of α10 subunits and effects of α9*-nAChR elevated function on EAE.

The current study demonstrates that there is attenuation of disease severity and delayed onset of disease symptoms in nAChR α9/α10 DKO mice, as was seen previously in α9 KO or in nicotine-treated mice [15,17,31,32]. However, protection again is only partial—whether for DKO animals in this study or for α9 KO or nicotine-exposed mice, there still is presence and evolution of EAE, albeit attenuated. On one hand, this suggests that mechanisms in addition to those controlled by α9*-nAChR play roles in the inflammatory and hyperimmune response. These processes could involve other nAChR subtypes or could be mediated by non-nicotinic signals.

Nevertheless, levels of disease attenuation are similar across or within studies for nicotine-treated [17,31], α9 KO [17], or α9/α10 DKO (this study) mice. This is true whether measured in absolute terms or when data are normalized to that seen for untreated WT or WT littermate controls, in which comparable, absolute levels of peak disease also were induced across all studies. Furthermore, the delay in onset of disease and in the time when peak disease severity is similar for nicotine-treated [17,31], α9 KO [17], or α9/α10 DKO (this study) mice. Since there is no difference between effects on EAE in α9 KO or α9/α10 DKO mice, it is possible that protective effects of α9*-nAChR genetic manipulation could be due to elimination of homomeric α9-nAChR function alone. Moreover, the lack of evident differences between effects of α9 KO and α9/α10 DKO minimizes the likelihood that α10 subunits, alone or in conjunction with α9 subunits, have roles in the disease process that would add to or circumvent influences due to loss alone of α9 subunits. This suggests that any elevation of α9*-nAChR ionotropic function due to α10 co-expression is non-consequential in the absence of α9 subunits or when α9- and α9/α10-nAChR function is blocked in the presence of nicotine. This is consistent with current perspectives that there are no functional, homomeric α10-nAChR. These findings also discount the possibility that nAChR α10 subunits contribute to alternate (e.g., metabotropic) functions in immune cells influencing the EAE response. If these were present, one would expect there to be differences in EAE presentation between α9 KO and α9/α10 DKO mice, but we see no such evidence. Studies of effects of nAChR α10 KO alone might be informative, but without evidence that they can assemble as homomers or as heteromeric partners with subunits other than α9 to form functional ion channels, and given our reasoning above about their contributions in metabotropic action, we do not see such work as being productive. However, importantly, and collectively with prior reports [15,17,31,32], the current findings add to earlier indications that α9*-nAChR significantly participate in inflammatory and autoimmune disease course and severity and that at least some aspects of nicotine’s protective effects can be tied to its antagonism of peripheral immune cell α9*-nAChR function.

Protection, albeit incomplete, also is seen in nAChR α9/α10 DKO animals against inflammatory cell infiltration into the brains and spinal cords of afflicted animals, and against ROS production in the brain. These effects reported here are observed when peak disease occurs in WT animals. The magnitude of the reduction in inflammatory cell infiltration at that time is slightly lower than the reduction observed in the spinal cords of nicotine-treated animals [31] assessed later in the disease, when animals had partially recovered, but this also is when the inflammatory response typically wanes. The magnitude of the partial reduction in ROS in the brains of α9/α10 DKO mice in this study is comparable to that seen in nicotine-treated animals at the same time [15,32]. Thus, these measures of EAE inflammation and autoimmunity are affected similarly by α9/α10 DKO or in nicotine-treated animals, where α9*-nAChR function would be antagonized.

Curiously, there is less myelin loss in the brains of WT or in α9/α10 DKO mice than in their spinal cords when measured 30 days after EAE induction and ~2 weeks after peak disease. Moreover, even though there is protection against earlier inflammation and ROS production in DKO animals, myelin loss in the brain is not diminished in DKO mice when measured another ~2 weeks later. This contrasts with the ~50% protection in α9/α10 DKO animals against the more elevated loss of myelin in the spinal cord as seen in this study. Nevertheless, this is comparable to the ~65% reduction in myelin loss in spinal cords from nicotine-treated animals at about the same time in the disease course [31]. Spinal lesions, defined by MRI and thus subject to specificity and sensitivity of imaging, often appear sooner or more prominently than brain lesions in human MS patients with primary progressive disease [54]. Perhaps similar phenomena are observed here in the mouse model, and further studies involving additional experimental conditions will address this possibility. It also is possible that demyelination in the brain is below a threshold level required for us to observe DKO protection.

There is very good agreement between our studies done using WT compared to α9 KO, nicotine-treated or α9/α10 DKO mice at different times and with different animal and reagent stocks [15,17,31,32] (this study). This agreement is not just qualitative, but also quantitative, in regard to effects on peak disease scores and disease time course, and on degrees of protection against demyelination, ROS production, and inflammatory infiltrates. One concern was that the doses of nicotine used in our earlier work, and that are limited by toxic effects if elevated too high, but are designed to mimic conditions in humans with regard to quasi-steady state blood levels, might not be high enough to completely antagonize α9*-nAChR function. However, the findings of our studies collectively that nicotine exposure neither adds to nor blunts effects of nAChR α9 KO, which are indistinguishable from those of α9/α10 DKO, indicate that nicotine treatment, at doses used, maximally blocks α9*-nAChR function as related to EAE development.

One way in which pro-inflammatory and autoimmune responses mediated by α9*-nAChR could be exacerbated is by elevated expression of heteromeric α9α10-nAChR relative to homomeric α9-nAChR with lower ionotropic function. However, close similarity between effects in the EAE model of α9 KO and of α9/α10 DKO would seem to dismiss this possibility, and although it is yet to be confirmed, peripheral immune cells involved are likely to naturally express the heteromer predominantly, assuming that they also express α10 subunits. Another possible means to elevate α9*-nAChR function, and perhaps to exacerbate EAE, is by introducing a GOF mutation into mice via a KI strategy. This is not literally an approach to provide an inverse of the DKO strategy, although it might be expected that EAE disease could be made worse by hyperactive α9*-nAChR responding to natural cholinergic signals. However, our studies do not show statistically-significant differences in EAE severity and rate or time of onset between WT littermates or nAChR α9 subunit GOF KI heterozygotic or homozygotic mice. The trend toward a slower recovery from peak disease in homozygotic or heterozygotic α9 KI mice might warrant future investigation, but we had expected that if there would be a GOF effect, it would be during the initiation and rise phases of the disease course, when the inflammatory response is most strongly manifest, rather than in the recovery phase, when inflammation is thought to dissipate. Nevertheless, and although the overall level of disease was quite low in our studies, we certainly do not see halving or doubling of effects of α9 KI as we do for α9 KO compared to WT mice. That notwithstanding, it is remarkable that disease severity is less than half of that in WT C57BL/6J mice than in WT animals on the FVB.129P2-*Pde6b^+^ Tyr^c-ch^*/AntJ background. This underscores the admonition that mouse genetic background matters in terms of responses to treatments and disease susceptibility. It may be of interest to determine whether inflammatory disease resistance is a feature of the FBV strain, whether introduction into those mice of the Pde6b allele to counter development of retinal degeneration and blindness in FBV mice also influences inflammatory and immune response in them, and whether that also is tied to cochlear degeneration and hearing loss.

Our earlier work suggested that nicotine exposure produces anergy in murine T cells emerging from interactions of hemopoietic stem cells with thymic cells in organ culture [13,16]. Functional inhibition by nicotine of α9*-nAChR well could account for these effects, consistent with other indications of T cell immaturity in α9 KO mice [32]. This idea is supported by findings that functional inhibition of α9*-nAChR has a modest effect on lymphocyte proliferation [43]. However, future studies should be interesting in terms of whether baseline levels of other immune system cells are altered in α9 KO, α9/α10 DKO, or α9 KI mice.

Interestingly, there is no evidence, admittedly in an unnatural vivarium setting, of immunodeficiency or of susceptibility to spontaneous infections in α9 KO or α9/α10 DKO mice. This indicates that there is not wanton destruction of the immune system in these animals, and it is consistent with attenuation, but not total elimination, of disease in the EAE model applied to α9 KO or α9/α10 DKO mice. Similarly, our studies using α9 KI mice are not consistent with animals displaying a hyperinflammatory phenotype, as they show no evidence for hyperimmunity or heightened infection susceptibility. Nevertheless, just as nicotine normalizes mood in a state-dependent manner, blunting depression but also blunting anxiety [55], nicotine acting through α9*- and perhaps other nAChR subtypes could moderate immune and inflammatory responses rather than push them to extremes.

Our findings are consistent with a variety of previous reports showing that exposure to nicotine dampens inflammatory responses in vivo [56,57,58]. On one hand, tobacco cigarette smoking is reported to exacerbate MS progression, symptoms, and risk [59,60,61]. However, snuff use has been associated with reduced susceptibility to MS [62,63], and we previously have pointed out that nicotine as a potential medicinal is very different from smoking [32], meaning that it or other antagonists of α9*-nAChR are viable therapeutic candidates.

## 5. Conclusions

Indications that nicotine can inhibit inflammation and be immunosuppressive are still relatively new, as are concepts about the so-called “cholinergic anti-inflammatory system” [64,65,66,67]. The current studies extend our intriguing findings implicating different nAChR subtypes in specific stages or events in inflammation and immunity that must be naturally responding to endogenous signaling via ACh [15,17,31,68,69]. Results of the current experiments again are consistent with the disease-exacerbating activity of α9*-nAChR (whether or not expressed with α10 subunits) in inflammatory and autoimmune responses. This means that ideas about roles of nAChR in inflammation require revision, because although some nAChR subtypes seem to have anti-inflammatory and immunosuppressive effects, α9*-nAChR play roles that are anything but anti-inflammatory. In fact, α9*-nAChR or their downstream mediators clearly are attractive targets for attenuation of inflammation and autoimmunity. This possibility is exciting because of the peripheral disposition of α9*-nAChR, not shielded from medicinals by the blood-brain barrier. Moreover, the unique pharmacology of α9*-nAChR suggests that they could be targeted by ligands not having effects at other nAChR subtypes. In fact, polypeptide α-conotoxins selective for α9*-nAChR already exist [70,71,72,73], have been characterized, and are being investigated for therapeutic potential based in part on reduction of immune system activity [74]. These approaches might be extended toward development of superior, auxiliary, or alternative approaches to block or control inflammation and immunity while having limited adverse side effect liability and abuse potential.

## Figures and Tables

**Figure 1 biomolecules-09-00827-f001:**
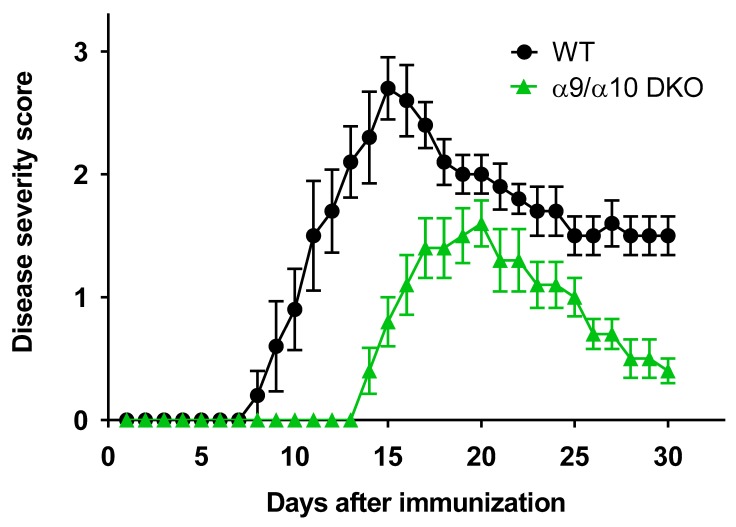
Attenuated experimental autoimmune encephalomyelitis (EAE) severity in α9/α10 DKO mice. EAE disease symptom evaluation was done for groups of wild-type (WT, ⬤) or nicotinic acetylcholine receptor (nAChR) α9/α10 subunit double knock-out (α9/α10 DKO, ▲) mice. Each data point represents the average score (+/− S.E.M.) for disease severity at the indicated time post-immunization for the specified mouse cohort. Reduced EAE severity was seen in α9/α10 DKO mice versus WT controls. N = 5 mice per group (note that another three animals having disease severity scores at day 14 representative of their groups were sacrificed for histological analysis at that point, but results for them up through day 14 are not included in the data shown). WT versus α9/α10 DKO for every time point from day 9 to day 30: *p* < 0.01, Mann–Whitney test.

**Figure 2 biomolecules-09-00827-f002:**
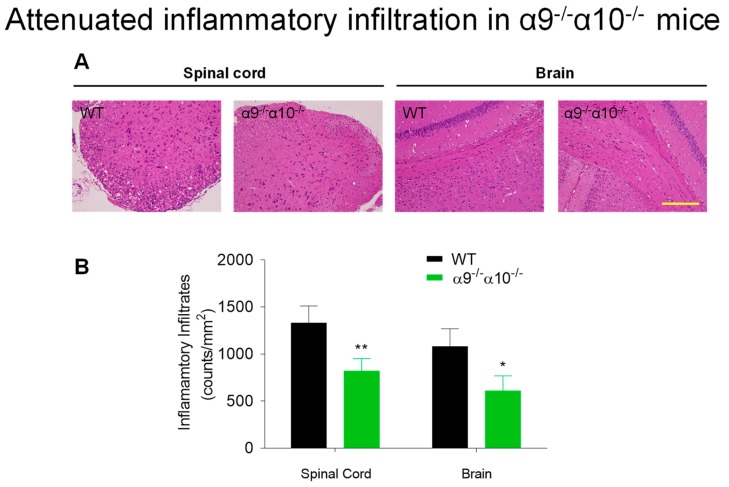
Attenuated inflammatory infiltration in EAE α9/α10 DKO mice. (**A**) Hematoxylin and eosin (H&E) staining reveals inflammatory cell infiltration in spinal cord or brain sections from wild-type (WT) or from nAChR α9/α10 subunit DKO mice (α9^−/−^α10^−/−^) at day 14 after immunization. Scale bar: 50 µm. (**B**) Quantified data show reduced inflammatory infiltrates in spinal cords and brains of α9^−/−^α10^−/−^ mice. N = 3 mice per group, four sections examined per mouse, n = 12. Mean ± S.E.M.; unpaired *t*-test; * *p* < 0.05, ** *p* < 0.01.

**Figure 3 biomolecules-09-00827-f003:**
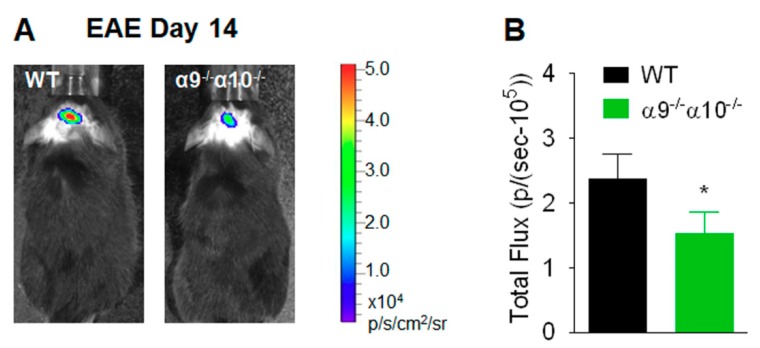
Reduced production of reactive oxygen species (ROS) in EAE α9/α10 DKO mice. (**A**) Visualization of ROS was done for wild-type (WT) or nAChR α9/α10 subunit DKO mice (α9^−/−^α10^−/−^) by in vivo bioluminescence imaging at day 14 after immunization. (**B**) Quantification of bioluminescence shows reduced signal intensity in α9^−/−^α10^−/−^ mice at day 14 after immunization. N = 5 mice per group. Mean ± S.E.M.; unpaired *t*-test; * *p* < 0.05.

**Figure 4 biomolecules-09-00827-f004:**
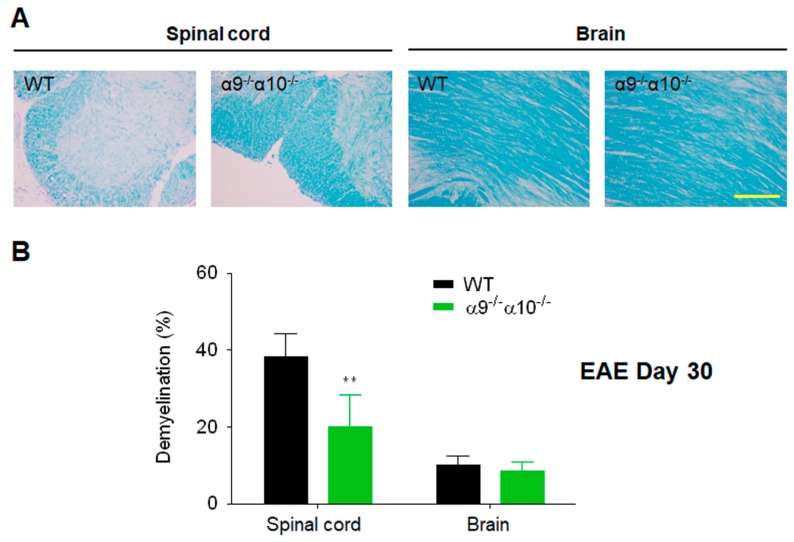
Reduced demyelination in EAE α9/α10 DKO mice. (**A**) Luxol blue staining reveals the intensity of myelin integrity in brain or spinal cord sections from wild-type (WT) or nAChR α9/α10 subunit DKO mice (α9^−/−^α10^−/−^) at day 30 after immunization. Scale bar: 50 µm. (**B**) Quantified data show reduced demyelination in spinal cords of α9/α10 DKO mice. N = 3 mice per group, at least four sections examined per mouse, n ≥ 12. Mean ± S.E.M.; unpaired *t*-test; ** *p* < 0.01.

**Figure 5 biomolecules-09-00827-f005:**
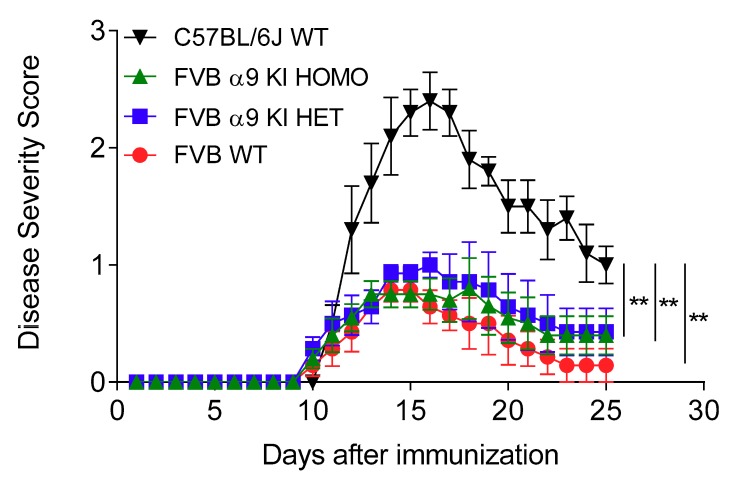
Effects of a nAChR α9 subunit gain-of-function (GOF) mutation on EAE severity. EAE disease symptom evaluation was done for groups of wild-type (WT) mice on the C57BL/6J background (C57BL/6J WT, ▼n = 5), or for nAChR α9 subunit knock-in heterozygotes (FVB α9 KI HET, ■, n = 7), homozygotes (FVB α9 KI HOMO, ▲, n = 10) or their WT littermate controls (FVB WT, ⬤, n = 7) on the FVB.129P2-*Pde6b^+^ Tyr^c-ch^*/AntJ (FVB) background. Reduced EAE severity was seen on FVB WT, FVB α9 KI HET, and FVB α9 KI HOMO mice as compared to C57BL/6J WT mice. Mean ± S.E.M.; Mann–Whitney test; ** *p* < 0.01 for the FVB cohorts relative to C57BL/6J WT mice.

**Table 1 biomolecules-09-00827-t001:** Disease parameters for EAE development in α9/α10 DKO mice.

	C57BL/6J WT	α9/α10 DKO	p (WT vs. DKO)
Mean EAE score (day 15)	2.7 ± 0.3	0.8 ± 0.2	0.0004
Peak disease severity	2.7 ± 0.3	1.6 ± 0.2	0.006
Cumulative disease score	39.3 ± 2.6	16.8 ± 2.3	0.0002
Incidence	5/5	5/5	-

Key parameters from the studies illustrated in Figure 1 are summarized. *P*-values are for one-way ANOVA comparing WT to α9/α10 DKO mice.

**Table 2 biomolecules-09-00827-t002:** Disease parameters for EAE development in α9 GOF KI mice.

	C57BL/6J WT	FBV WT	FBV α9 KI HET	FBV α9 KI HOMO
Mean EAE score (day 15)	2.3 ± 0.2	0.8 ± 0.1	0.9 ± 0.1	0.8 ± 0.1
*p* versus C57BL/6J WT		0.00002	0.00002	0.00001
Peak disease severity	2.4 ± 0.2	0.9 ± 0.1	1.0 ± 0.1	0.8 ± 0.3
*p* versus C57BL/6J WT		0.0005	0.005	0.001
Cumulative disease score	24.1 ± 1.2	6.6 ± 2.0	10.4 ± 2.9	9.0 ± 1.9
*p* versus C57BL/6J WT		0.00007	0.004	0.002
Incidence	5/5	7/7	10/10	5/5

Key parameters from the studies illustrated in Figure 5 are summarized. *P*-values are for one-way ANOVA comparing C57BL/6J WT mice to any of the FBV cohorts.

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
