# Peer review of "Attenuation in Nicotinic Acetylcholine Receptor α9 and α10 Subunit Double Knock-Out Mice of Experimental Autoimmune Encephalomyelitis"

_biomolecules, 2019, doi:10.3390/biom9120827_

Round 1
Reviewer 1 Report
In the study of Liu et al. was found that features of EAE (demyelination, oxidative stress, infiltration) could be attenuated in nAChR alpha9/10 subunit DKO mice. Furthermore, this effect is the same as previously revealed for nicotine administration or only alpha9 subunit KO. It is interesting study but for me it should be improved by addition the third group - alpha10 KO for all experiments.
Other concerns:
Lines 32-60: I propose that the Introduction should be more focused on alpha9/10 without common phrases.
Line 68: Please clarify why nicotine was named as antagonist. For me, I suggest to avoid such generalization. KO of alpha 9 can lead to increase expression other subunits of nAChR having anti-inflammatory properties, thereby enhancing effect of nicotine.
Line 81: What is “inotropic function”? nAChR is an inotropic receptor type.
Line 84: Please see https://www.pnas.org/content/109/11/4308. Permeability of alpha9/10 nAChRs for calcium could be different. Also these receptors could be expressed in mitochondria. Please see: https://www.ncbi.nlm.nih.gov/pmc/articles/PMC5601054/
Please indicate in the Introduction that α9α10 nAChR is analgesic target for treatment of neuropathic pains
Fig 1 and 5: Please indicate changes in each of parameters that together constitute the score. I saw lines 150-151, but I suggest that readers should know the changes in limb weakness/paralysis, tail stiffness, and animal health. What is «animal health»? How it can be quantified?
Fig, 2: To estimate inflammatory reactions, it will be good to additionally detect changes in inflammatory cytokines (e.g., TNFa, IL-1, IL-6, IL-17)?
Fig 3. There is no ideal ROS sensor, so I think that including antioxidant treated group for luminol experiments will be helpful.
Author Response
In the study of Liu et al. was found that features of EAE (demyelination, oxidative stress, infiltration) could be attenuated in nAChR alpha9/10 subunit DKO mice. Furthermore, this effect is the same as previously revealed for nicotine administration or only alpha9 subunit KO. It is interesting study but for me it should be improved by addition the third group - alpha10 KO for all experiments.
RESPONSE: We appreciate this reviewer’s perspective and curiosity, and we recognize that the study under review still leaves a formal question about effects of nicotinic acetylcholine receptor (nAChR) α10 subunit deletion alone on the experimental autoimmune encephalomyelitis (EAE) response. In fact, we acknowledge that in the Discussion’s third paragraph of the manuscript (“Studies of effects of nAChR α10 KO alone might be informative, but without evidence that they can assemble as homomers or as heteromeric partners with subunits other than α9 to form functional ion channels, and given our reasoning above about their contributions in metabotropic action, we do not see such work as being productive.“). However, because our results offer not even a hint, let alone compelling evidence, for an effect of α10 subunits alone, we think that conducting an additional set of studies using nAChR α10 subunit knock-out mice would be a futile exercise and not worth the investment of our time and resources. This is in addition to knowledge that α10 subunits in mammals evolved in parallel to the prestin protein found in the cochlea, conferring the then new α9α10-nAChR subtype with a unique function beyond that of homomeric α9-nAChR. Extensive study of α9α10-nAChR in the cochlea has revealed only a minor effect (at best) following the deletion only of α10 subunits. The likelihood is that α10 KO animals will have WT mouse-like responses in the EAE model. Reviewer 2 shares our perspective.
Other concerns:
Lines 32-60: I propose that the Introduction should be more focused on alpha9/10 without common phrases.
RESPONSE: We appreciate this suggestion, but we think that the first three paragraphs are useful if not required for readers in a presumably diverse audience to have some of the critical background information about nicotinic receptors, acetylcholine’s roles in the body, and the “yin-and-yang” of inflammatory responses. However, we will concede to the editor’s perspective on this matter.
Line 68: Please clarify why nicotine was named as antagonist. For me, I suggest to avoid such generalization. KO of alpha 9 can lead to increase expression other subunits of nAChR having anti-inflammatory properties, thereby enhancing effect of nicotine.
RESPONSE: In keeping with our intent to provide critical background information to readers from a diverse audience, and by using universally-accepted terminology, we point out the simple fact that nicotine is an antagonist at α9*-nAChR, by contrast to nicotine’s effects as an agonist at other nAChR subtypes. That is, at the receptor level, nicotine blocks (i.e., antagonizes) rather than mimics the effects of acetylcholine as an agonist stimulating function of α9*-nAChR. Nonetheless, we have clarified in the revised manuscript that nicotine has that atypical action for α9*-nAChR having ionotropic receptor function.
Line 81: What is “inotropic function”? nAChR is an inotropic receptor type.
RESPONSE: Pertinent to our intention to provide appropriate background information to a diverse audience, we point out in the first paragraph and again in the fourth paragraph of the Introduction that nAChR are viewed as ligand-gated ion channels (ionotropic neurotransmitter receptors).
Line 84: Please see https://www.pnas.org/content/109/11/4308. Permeability of alpha9/10 nAChRs for calcium could be different. Also these receptors could be expressed in mitochondria. Please see: https://www.ncbi.nlm.nih.gov/pmc/articles/PMC5601054/
Please indicate in the Introduction that α9α10 nAChR is analgesic target for treatment of neuropathic pains
RESPONSE: We indicate that α9*-nAChR are Ca2+ permeable in the fifth paragraph of the Introduction. Some of us have expressed our concerns (Ref. 34; Morley, B. J.; Whiteaker, P.; Elgoyhen, A. B., Commentary: Nicotinic Acetylcholine Receptor α9 and α10 Subunits Are Expressed in the Brain of Mice. Front Cell Neurosci 2018, 12, 104.) about the data in the second citation mentioned, and we thus have chosen not to cite it. The last two references in this manuscript refer to potential involvement of α9*-nAChR in inflammatory pain.
Fig 1 and 5: Please indicate changes in each of parameters that together constitute the score. I saw lines 150-151, but I suggest that readers should know the changes in limb weakness/paralysis, tail stiffness, and animal health. What is «animal health»? How it can be quantified?
RESPONSE: We have added more specifics under the EAE section of Materials and Methods about disease scoring. “Animal health” means survival and ability to thrive; a score of 5 is given for moribund or deceased animals, and a score of 4 means that we may need to engage in compassionate sacrifice.
Fig, 2: To estimate inflammatory reactions, it will be good to additionally detect changes in inflammatory cytokines (e.g., TNFa, IL-1, IL-6, IL-17)?
RESPONSE: We agree, and we have done so in some of our other studies, showing expected changes in pro- or counter-inflammatory cytokines based on and proportional to disease severity and other indices, but we did not include such assays in this study.
Fig 3. There is no ideal ROS sensor, so I think that including antioxidant treated group for luminol experiments will be helpful.
RESPONSE: Our intention was to assess whether there is a reduction in ROS in double KO mice relative to wild-type animals, all afflicted with EAE, and we think that our data demonstrate such.
Reviewer 2 Report
This work is a continuation of previous studies led by R.J. Lukas regarding the roles of α9 and α9α10 nAChRs in non-CNS cells, where there is consent among the majority of researchers that α9*-nAChRs are mainly expressed. The authors study the effects of: i) α9/α10 DKO in mice and ii) an α9 gain-of-function mutant KI in mice, with regard to pathophysiological situations (ROS species, inflammatory cell infiltration, demyelination) observed in experimental autoimmune encephalomyelitis (EAE) and compare their findings with their previous studies.
As a general remark, this paper is another valuable contribution to the development of α9α10 nAChR as a potential pharmaceutical target.
The experimental section is described adequately clear and depicts the cautiousness by which the authors have conducted experiments diminishing subjectivity. Similarly, the results are described with high clarity, with minor exceptions which are described below.
The introduction is comprehensive and introduces concisely issues related to the function of a9*-nAChRs in immune cells e.g. possible effects of ionotropic function, the possibility to display metabotropic function etc.
Minor issues:
At page 2, line 69 the authors mention “… at phylogenetically-ancient α9* nAChR”. The phrase is kind of distracting to the readers. I am not aware of any study focusing on naturally occurring haplotypes of α9 nAChR encoding distinct alleles which shows difference in functionality (e.g. ligand binding or electrophysiological properties). Even if there is such a study, in the KI experiments which α9 do the authors use? I suggest removal of “phylogenetically-ancient”. Although it is probably out of the scopes of the authors, the fact that the disease severity, delay in disease onset etc are indistinguishable among the α9 only KO, the nicotine treated and the α9α10 DKO mice, inductively provides information about the ligand binding sites present in α9* nAChRs in immune cells. Could their studies imply a preferable α9/α10 stoichiometry expressed in immune cells? I find the text from line 311 to 317 confusing regarding the initial plans of the authors in the present study and/or in conjunction with their previous studies. The "α9 KI WT" label in Figure 5 is confusing and should be changed. KI and WT together? Instead, the FVB background should be noted.
Weaknesses:
The authors have not conducted RNA-seq or immunostaining to verify transcription and surface expression of the α10 nAChR subunit, despite the fact that the study is based on the assumption (stated as such at line 423) that the immune cells express predominantly α9α10 heteropentamers and not α9 homopentamers. In the Discussion section they explain delicately the reasoning for not pursuing experiments with α10 knock-out mice (I agree there is no evidence for the existence of α10 homopentameric nAChR functional ion channels and moreover possible metabotropic function would have an impact in α9/α10 DKO mice), but nevertheless, due to the lack of experiments with α10 nAChR the possibility that the protective effects are due to α9 alone and not α9α10 heteropentamers cannot be precluded. The choice of using FVB background mice in the α9 KI experiments does not confer any statistically significant results regarding the EAE disease course and severity and therefore cannot address issues regarding the ionotropic vs metabotropic function of α9*nAChR in immune cells. In this regard, I find the text from line 311 to 317 confusing.
I find as an ultimate proof of the α9*-nAChRs involvement in immune cells function the incorporation of a loss-of-function mutation/KI (e.g. mutation of the loopB Trp residue), which could also provide insights about the mode of action (orthosteric vs allosteric or even ionotropic vs metabotropic).
Round 2
Reviewer 1 Report
I appreciate comments of authors and efforts to do the manuscript better. In general, I have been satisfied the revised version.